# Harmonization of Meteosat First and Second Generation Datasets for Fog and Low Stratus Studies

**Sheetabh Gaurav \***, **Sebastian Egli**, **Boris Thies and Jörg Bendix**

Laboratory for Climatology and Remote Sensing (LCRS), Department of Geography, University of Marburg, Deutschhausstr. 12, 35032 Marburg, Germany; eglis@staff.uni-marburg.de (S.E.); thies@staff.uni-marburg.de (B.T.); bendix@geo.uni-marburg.de (J.B.)
**\*** Correspondence: gaurav@staff.uni-marburg.de; Tel.: +49-6421-2824204

**Abstract:** Operational weather satellites, dating back to 1970s, currently provide the best basis for climatological investigations, such as an analysis of changes in the cloud cover. Because clouds are highly dynamic in time, temporally high-resolution data from the geostationary orbit are preferred in order to take variations in the diurnal cycles into account. For such studies, a consistent dataset in space and time is mandatory, but not yet available. Ground-based point measurements of various cloud parameters, such as ceiling, visibility, and cloud type are often sparsely spread and inconsistent, making it difficult to derive reliable spatio-temporal information over large areas. The Meteosat program has generally provided suitable data from over Europe since 1977, but different spatial, spectral, and radiometric resolution of the instruments of the individual satellites, including early-years calibration uncertainties, makes harmonization necessary to finally derive a time series applicable to any kind of climatological study. In this study, a machine learning-based approach has been employed to generate a long-term consistent dataset with high spatio-temporal resolution and extensive coverage over Europe by the harmonization of Meteosat First Generation (MFG) and Meteosat Second Generation (MSG) satellite datasets (1991–2020). A random forest (RF) regressor is trained on the overlap period (2004–2006), where datasets of both satellite generation (MFG and MSG) are available to predict MFG Water Vapour (WV) and Infrared (IR) channels brightness temperature (BT) values based on MSG channels. The aim of the study is to synthesize MFG MVIRI data from MSG SEVIRI to generate a consistent MFG time series. The results indicate a good match of MFG synthesized data with the original MFG data with a mean absolute error of 0.7 K for the WV model and 1.6 K for the IR model, and an out-of-bag (OOB) R² score of 0.98 for both the models. Based on the trained models, the MFG scenes are synthesized from the MSG scenes for the years from 2006 to 2020. The long-term homogeneity of the generated time series is analyzed. The harmonized dataset will be applied to generate a continuous time series on fog and low stratus (FLS) occurrence for a climatological time scale of 30 years.

**Keywords:** climatology; fog; low stratus; remote sensing; Meteosat First Generation; Meteosat Second Generation; harmonization; random forest



## 1. Introduction

Geostationary satellite systems, such as the Meteosat program, have been collecting enormous amounts of data and disseminating helpful information about the atmosphere and the Earth's surface since the 1970s. Geostationary satellite datasets with high resolution in both time and space can compensate for the disadvantage of discontinuous ground-station measurements in order to provide valuable and meaningful information that can be utilized for climate studies at the continental and regional scale [1–3]. They have the advantage of frequently viewing the same large area of the Earth's atmosphere allowing for temporal continuity in the observation [4]. With the Meteosat program by European Space Agency (ESA) and European Organisation for the Exploitation of Meteorological

Satellites (EUMETSAT), different geophysical parameters, such as concentrations of major atmospheric components (such as carbon dioxide, ozone, etc.), precipitation, cloud systems, winds, and many other parameters can be detected, retrieved, or monitored [5–8].

For climatological studies, a long-term spatio-temporally harmonized dataset is needed which is suitable for the analysis of climate variability and the detection of climate trends. These datasets can also be used to validate global climate model simulations. However, due to changes in the sensor characteristics of the satellite systems, generating harmonized datasets from multiple satellites over many decades becomes a challenging task [9]. The term "harmonized" here means that the differences between the instruments of different generation satellites are removed by inter-calibration with the final aim of obtaining a synthetic one instrument dataset over the entire time series.

The Meteosat program has existed since the 1970s and could, therefore, be used for long-term observations. The Meteosat Visible and InfraRed Imager (MVIRI) on-board the MFG satellites has been operating from 1977 to 2006, and the Spinning Enhanced Visible and InfraRed Imager (SEVIRI) has been operating on-board the MSG satellites from 2004 onwards. However, the difference in the sensor equipment of the satellites, as well as due to the sensor degradation over its lifetime, may lead to inhomogeneities in data evaluation which must be corrected in advance of climatological time series analyses [10,11]. Recently, EUMETSAT released the Fundamental Climate Data Record (FCDR) of recalibrated Level 1.5 thermal Infrared (IR), Water Vapour (WV), and Visible (VIS) radiances from the MVIRI instrument onboard the MFG satellites from the year 1982 onwards [12,13]. This extends the available weather satellite to a climatological scale of more than 30 years. The MVIRI FCDR dataset is a harmonized data record of multiple sensors (MFG 2–7) to correct for inter-satellite differences with each sensor being calibrated and adjusted to sensor degradation ensuring accuracy and stability in both space and time. The consistent recalibration improved spectral response function (SRF) characterization and improved quality flagging resulted in significant improvements in MVIRI measurements, particularly for high clouds [12]. The calibration of MVIRI WV and IR channels was performed by EUMETSAT using the High-Resolution Infrared Radiation Sounder (HIRS) instrument on board the National Oceanic and Atmospheric Administration (NOAA) polar orbiting platforms [13].

Previous studies have shown that a linear combination of well correlated narrow bands can be used to synthesize the broadband radiances in the same wavelengths [14–16]. Cros et al. [15] have shown a linear relationship between the reflectances of SEVIRI narrow bands, i.e., 0.6 and 0.8 µm channels and the MVIRI broadband visible channel due to the spectral overlap of these channels. Following this approach, Posselt et al. [17] extended the Satellite Application Facility of Climate Monitoring (CM SAF) surface radiation climatology, earlier derived from MFG satellites (1983–2005) [18], up to 2010 using the measurements from the MSG satellites. The retrieval method of CM SAF Top of Atmosphere (TOA) Radiation long term data record used the MVIRI/SEVIRI overlap period (2004–2006) to derive the empirical narrowband to broadband regressions [19]. The EUMETSAT CM SAF released the Cloud Fractional Cover dataset from Meteosat First and Second Generation (COMET) covering 1991–2015 at 0.05 degree spatial and 30 min temporal resolution to provide accurate estimates of monthly diurnal cycle by calculating the Cloud Fractional Cover (CFC) [20]. The COMET retrieval algorithm used the Meteosat heritage channels to ensure consistency between retrievals from both the sensors, which had already been inter-calibrated. The broadband VIS channel was simulated by use of the linear combination of reflectances of the two narrow-band 0.6 and 0.8 µm MSG SEVIRI channels. For MVIRI IR channel (10.5–12.5 µm) channel, SEVIRI's 10.8 µm IR channel (9.8–11.8 µm) was used to closely replicate the Meteosat heritage IR channel [21].

For the case of climatological FLS studies, it is essential to have a long-term spatially extensive time series (30 years or more) on FLS distribution. Unfortunately, there is no such methodology at present which can derive a consistent FLS product (day and night, 30 min resolution) for Europe using MFG and MSG for such a long time period. For

this purpose, due to the differences in the sensor characteristics of MVIRI and SEVIRI, a new harmonization scheme needs to be developed based on cross-calibration of MFG and MSG dataset. Most of the previous Meteosat-based harmonization studies have so far focussed on simulating the broadband channel similar to MFG MVIRI VIS channel from the two narrow-band visible channels of MSG SEVIRI based on their desired applications. To have a consistent FLS product, it is very important to have a long-term harmonized dataset in WV and IR channels because of their 24 h availability and significant role in FLS studies. Egli et al. [22] showed the significant contribution of SEVIRI 6.2 μm and 7.3 μm WV channels for the estimation of cloud base altitudes (CBA), which was further used to derive regions with fog occurrences. Despite the fact the WV channels are not sensitive to lower atmosphere, they are able to capture upper atmospheric circulation patterns and water vapour content which could aid in determining FLS. The introduction of MVIRI FCDR by EUMETSAT which guarantees better stability of MVIRI datasets [13] opens up the possibility of better harmonization results. Among the cited harmonization studies in the previous paragraph, only the COMET CFC retrieval algorithm [21] has employed MVIRI FCDR previously but they have synthesized just the IR channel apart from the VIS channel, by simply replicating the MSG SEVIRI IR 10.8 μm channel. According to the performance assessment study of the COMET CFC dataset [23], it underestimates the low stratus clouds occurring during the winter time over the continental Europe. This underlines the importance of having long-term harmonized datasets in WV and IR channels which can be used to develop an algorithm suitable for FLS classification and generating a long-term time series of FLS distribution.

In this study, an attempt has been made to generate a long-term spatially extensive consistent dataset (30 years) based on the harmonization of MFG and MSG datasets using a machine learning-based model, which can be used for FLS climatological studies. This MFG and MSG harmonization scheme was applied over the European domain (WMO region VI). In order to ensure the consistency of the dataset to be generated, it is necessary to first adapt the MSG dataset as per the specifications of the MFG dataset. The inter-calibration of MFG and MSG datasets was completed based on training a machine learning based model during the overlap period 2004–2006, since during this period both satellite series provide data and a direct comparison between the model results on the basis of both MFG and MSG datasets is possible. The trained models were used to synthesize the MFG MVIRI WV and IR channels from the year 2006 to 2020 using the MSG SEVIRI WV and IR channels in order to generate a long-term dataset spanning 30 years. An important point to note here is that in this study we are downgrading MSG datasets to MFG datasets in order to obtain a long-term time series for FLS climatological studies and not vice versa. This is due to the limited number of MFG channels available at our disposal to simulate the MSG channels and the significant emissivity differences between 3.9 μm and 10.8 μm IR channel which restricts the synthesis of an important FLS discriminating 3.9 μm channel from MFG IR channel. The only reason for upgrading MFG to MSG for FLS studies would be the synthesis of the 3.9 μm channel which is essential for FLS detection. However, the detection is based on a low spectral emissivity of FLS in this channel [24,25]. Because the 10.8 μm channel is not affected by this reduction in emissivity, a proper simulation of the 3.9 μm channel with MFG is, therefore, not possible.

The paper is structured as follows. In Section 2, an overview of MFG and MSG satellite datasets is given along with the detailed methodology for harmonization of the two datasets. In Section 3, the experimental results obtained are reported along with the interpretation and discussion of the results. Finally, a summary and conclusions are presented in Section 4.

## 2. Data and Methods

In this section, first a general overview about MFG and MSG datasets is given. Then the MFG and MSG data preprocessing steps are described followed by a brief description about the spatial and temporal resampling approach. After that, we describe the machine learning (ML) model generation for the inter-calibration of the two datasets and the final synthesized MFG dataset in detail (see Figure 1).

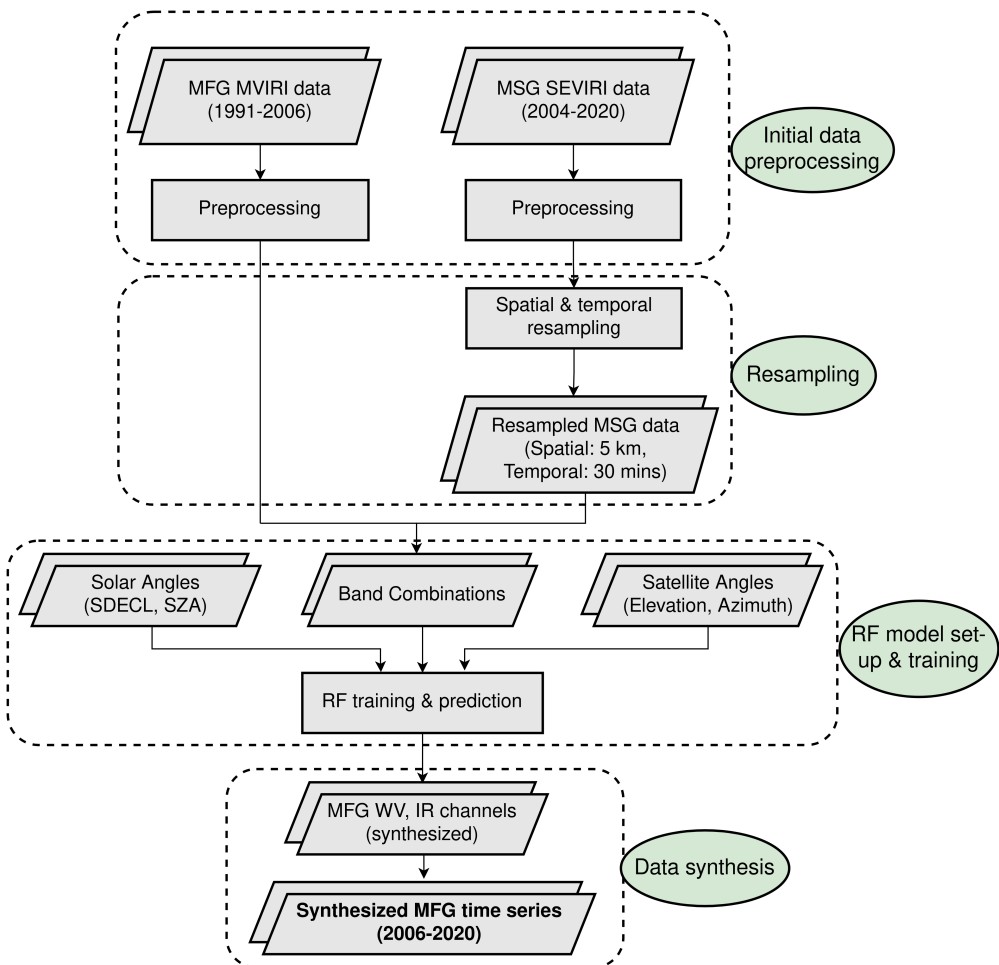

**Figure 1.** Flowchart depicting the methodology followed for the inter–calibration of MFG and MSG datasets. (SDECL: Sun Declination, SZA: Solar Zenith Angle, and RF: Random Forest).

### 2.1. Meteosat First and Second Generation Data

The MFG satellites used MVIRI, a radiometer scanning the Earth every 30 min in three spectral channels (known as the Meteosat heritage channels) covering visible and infrared wavelengths: the broadband visible channel (VIS, 0.5–0.9 μm), the water-vapor absorption channel (WV, 5.7–7.1 μm), and the thermal infrared channel (IR, 10.5–12.5 μm). The continuous archive data are available starting from the year 1982 onwards, despite the first satellite (MFG-1) being launched in 1977. The MVIRI instruments use a reflecting telescope to measure radiation in three spectral channels, i.e., the VIS, WV, and IR channels at a spatial resolution of 2.5 km (VIS) and 5 km (WV and IR). The MFG series consist of seven spin-stabilized geostationary satellites, named Meteosat-1 to -7. The MVIRI Level 1.5 data were retrieved from the EUMETSAT Data Center as images of 5000 × 5000 pixels (VIS) and of 2500 × 2500 pixels (WV and IR). Except for the VIS images of Meteosat-2 and -3, which only employ 6 bits, the values are encoded on 8-bits. The Level 1.5 data are built from the original Level 1.0 data by correcting them for undesirable geometric effects and by rectifying them on a reference geostationary grid. The MVIRI FCDR used in this study now contains reconstructed spectral response functions (SRFs) that account for the spectral

degradation of the sensors [26]. Those SRFs are used for a consistent recalibration of all observations, spanning MVIRI on Meteosat-2 to -7.

The SEVIRI aboard the MSG satellites is designed for the continuous monitoring of the Earth–atmosphere system. At a repeat rate of 15 min, data are collected in 11 infrared and visible spectral channels from 0.56 to 14.4 µm at sub-satellite resolution of 3 km and one high resolution visible (HRV) channel with a resolution of 1 km [27]. All recorded scenes of 3712 × 3712 pixels between 2004 and 2020 were acquired from EUMETSAT in a High Rate Information Transmission (HRIT), Network Common Data Form (NetCDF), and Hierarchical Data Format (HDF) Level 1.5 format. These data have been geolocated and corrected for all radiometric and geometric effects, making them suitable for the derivation of meteorological products.

### 2.2. Data Preprocessing

The satellite data (MFG and MSG) were preprocessed for the later processing in the machine learning method. They were brought to a uniform format (NetCDF) and tailored to the study area of interest (Figure 2).

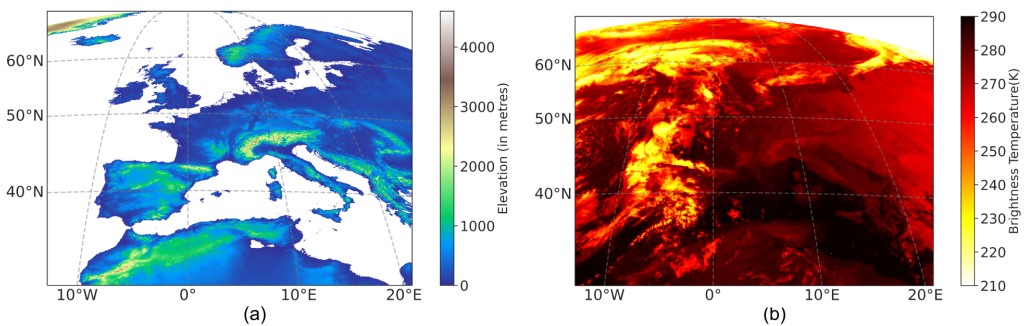

**Figure 2.** Region of interest for the inter-calibration of Meteosat datasets (**a**) DEM (Source: WorldClim DEM [28]), and (**b**) an exemplary MFG scene IR channel—30 October 2005, 00:00–00:30.

### 2.2.1. MFG Data Preprocessing

The Level 1.5 data of the MFG series were transmitted by EUMETSAT for the entire period (1991–2006). The harmonization and homogenisation of the dataset (WV and IR channels) was completed by applying the conversion factors and the spectral band adjustment factors. The WV/IR channel was provided as raw count values with corresponding re-calibration coefficients provided with the metadata of the scenes. The WV and IR radiances in mWm$^{-2}$sr$^{-1}$cm$^{-1}$ were derived from the given WV and IR counts ($c_{WV}$ and $c_{IR}$) with Equations (1) and (2), respectively:

$$L_{WV} = a_{WV} + b_{WV} * c_{WV} \qquad (1)$$

$$L_{IR} = a_{IR} + b_{IR} * c_{IR} \qquad (2)$$

where $a_{WV}$ and $a_{IR}$ are the offsets and $b_{WV}$ and $b_{IR}$ are the slope parameters. The radiance values of WV and IR channels obtained from Equations (1) and (2) were used to calculate the brightness temperature (BT) of WV and IR channels (Equations (3) and (4)):

$$BT_{WV} = n_{WV}/log(L_{WV}) - m_{WV} \qquad (3)$$

$$BT_{IR} = n_{IR}/log(L_{IR}) - m_{IR} \qquad (4)$$

where $n_{WV}$ and $n_{IR}$ are the radiance to BT conversion offsets and $m_{WV}$ and $m_{IR}$ are the radiance to BT conversion slopes of WV and IR channels respectively [26]. The quality flag masks were also included by setting the pixels flagged as bad quality as no value. The final preprocessed scenes were saved in NetCDF file format.

### 2.2.2. MSG Data Preprocessing

The Level 1.5 data, derived from the Level 1.0 data that were acquired by the MSG satellite, were provided by EUMETSAT for the entire study period (2004–2020). The original scenes were delivered in HRIT, NetCDF, or HDF format which were finally brought to the common NetCDF file format. The original calibration coefficients included in the Level 1.5 files from EUMETSAT were used for MSG SEVIRI calibration. The satellite data preprocessing of these scenes in different file formats were carried out by the Python-based package, Satpy [29].

### 2.3. Selection of MSG Channels Combination for the Input Training Data

For the inter-calibration of MFG and MSG data to obtain a consistent dataset over a time period of 1991–2020, ML-based models were trained on the overlap period (2004–2006), where both generation (MFG and MSG) satellite datasets are available (see Figure 3). The models were trained based on the MSG channels along with other features to predict the BT values of MFG WV and IR channels. For this, the selection of the right MSG channels combination was very important. The MSG channels were selected on the basis of the spectral overlap with the MFG channels. For the MFG WV channel, a significant overlap can be observed with the MSG WV062 channel and a small overlap with the MSG WV073 channel (see Figure 4). Similarly, for the MSG IR channel, there is a substantial overlap with the MSG IR108 and MSG IR120 channels, and some overlap with the MSG IR134 channel (see Figure 4). Based on this information, MSG WV062 and MSG WV073 channels were selected as the predictors for the model to synthesize the MFG WV channel. A combination of the channels MSG IR108, MSG IR120, and MSG IR134 were deemed suitable to synthesize the MFG IR channel.

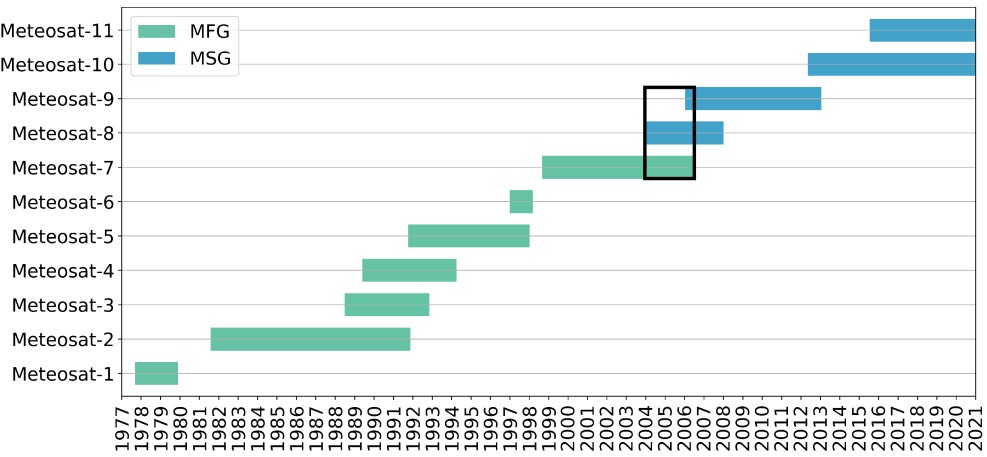

**Figure 3.** Timeline of different series of Meteosat satellites. The black rectangular box highlights the overlap period of MFG and MSG (2004–2006).

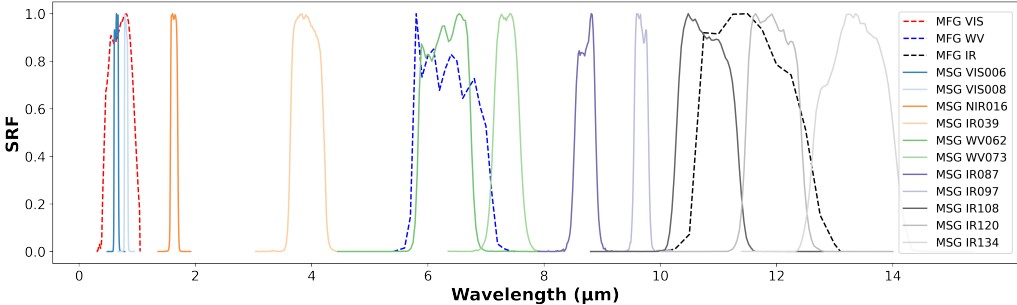

**Figure 4.** Spectral response curves for Meteosat-7 (MFG) and Meteosat-8 (MSG).

### 2.4. Spatial and Temporal Resampling of MSG Data

In order to achieve homogeneity of the final dataset, it was important to have the two datasets in the same spatial and temporal resolution. Since our task in this study was to train the ML model to predict the MFG channels from the MSG channels and to ultimately synthesize the MFG dataset for the time period 2006–2020, first the MSG datasets were reduced to the spatial (5 km) and temporal (30 min) resolution of the MFG dataset.

For spatial resampling, the nearest neighbor interpolation method was used. All the MSG channels corresponding to the MFG WV/IR channels, i.e., MSG WV062, WV073, IR108, IR120, and IR134 were resampled to the spatial resolution of the MFG WV/IR channels (5 km).

The SEVIRI onboard MSG satellites scans the Earth every 15 min as compared to MVIRI onboard MFG satellites which scans at every 30 min. For example, there are two MSG scenes (09:00–09:15 and 09:15–09:30) corresponding to a single MFG scene (09:00–09:30). So, the two MSG scenes corresponding to a MFG scene needed to be combined to bring it to the same temporal resolution as that of the MFG scene. The method adopted to combine the two MSG scenes was considered on the basis of the scan times of the two subsequent MSG scenes and a MFG scene. The scenes were combined by linearly blending the scan time difference of the two MSG scenes with respect to the MFG scene thus eliminating the scan time difference with the MFG scene. This was performed in the following steps:

- A reference timestamp of the middle pixel of the MFG scene was chosen and the scan time difference of each pixel in the MFG scene ($\Delta t\_mfg$) was calculated with respect to the reference timestamp.
- The time difference of each pixel of the two MSG scenes ($\Delta t\_msg1$ and $\Delta t\_msg2$) was calculated with respect to the reference timestamp defined in the previous step.
- The MSG time differences were resampled to the resolution of the MFG scene using nearest neighbor interpolation.
- The respective ratios (r1 and r2) for each pixel with latitude(y) and longitude(x) were calculated as per the following equations:

$$r1(y,x) = 1 - (\Delta t\_mfg(y,x) - \Delta t\_msg1(y,x))/(\Delta t\_msg2(y,x) - \Delta t\_msg1(y,x)) \tag{5}$$

$$r2(y,x) = 1 - (\Delta t\_msg2(y,x) - \Delta t\_mfg(y,x))/(\Delta t\_msg2(y,x) - \Delta t\_msg1(y,x)) \tag{6}$$

- The two MSG scenes were finally combined using the ratios calculated in Equations (5) and (6) as follows:

$$combined\_scene(y,x) = (msg\_scene1(y,x) * r1(y,x)) + (msg\_scene2(y,x) * r2(y,x)) \tag{7}$$

where *msg_scene*1 and *msg_scene*2 are the arrays with BT values of MSG scene 1 (09:00–09:15) and MSG scene 2 (09:15–09:30), respectively. *combined_scene* is the final combined MSG scene to bring it in the same temporal resolution as that of the MFG scene (09:00–09:30).

The scan time of the MFG scene does not exactly lie in the middle of the two MSG scenes, which is why this method was chosen to combine the two MSG scenes rather than simply taking the mean BT values of the two (see Figure 5). For example, for temporally resampled MSG scene corresponding to 1 January 2006 (09:00–09:30) original MFG scene for the European domain, linearly blended MSG scenes based on exact scan time differences yield a better result with a mean absolute shift ($\Delta$) of 2.52 K and standard deviation ($\sigma$) of 1.89 K as compared to the MSG scenes combined using mean BT values with $\Delta$ and $\sigma$ values of 2.63 K and 2.12 K, respectively (see Figure 6). This is evident from the histogram analysis which shows higher density of pixels close to the BT shift of 0 K in the case of MSG scenes combined by the method described above (Figure 6d). Apart from checking the two method of combining MSG scenes as described above, another check was performed to

check whether our selected method of combining the two MSG scenes results in a lower difference between MFG and MSG channels as compared to just using the nearest MSG scene. Please refer to Figure A1 in the Appendix A for detailed comparison.

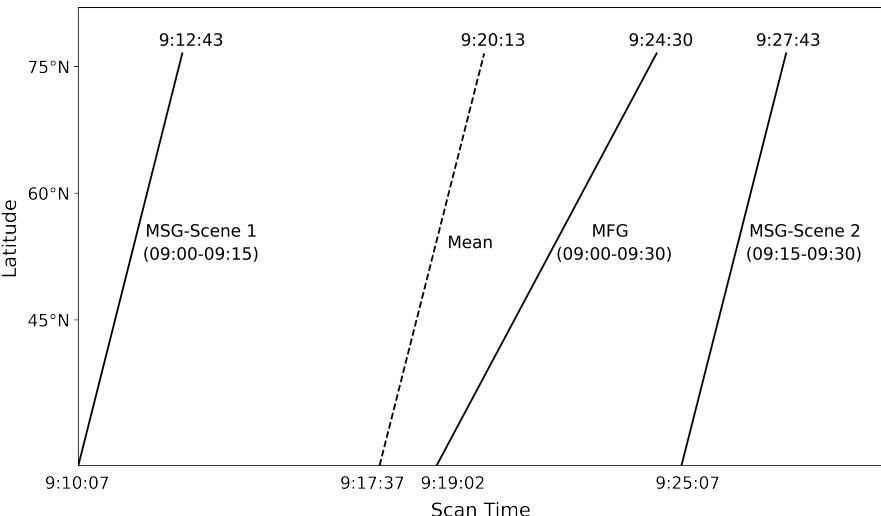

**Figure 5.** Plot depicting the scan times of one MFG slot (e.g., 09:00–09:30) and the two enveloping MSG scenes (e.g., 09:00–09:15, 09:15–09:30) along the latitudinal gradient of our study area. The dashed line represents the arithmetic mean of the two MSG scan times which is not fitting the MFG time. To better reproduce the temporal coverage of MFG with the two MSG scenes, we introduced weighted averaging (Equations (5)–(7)).

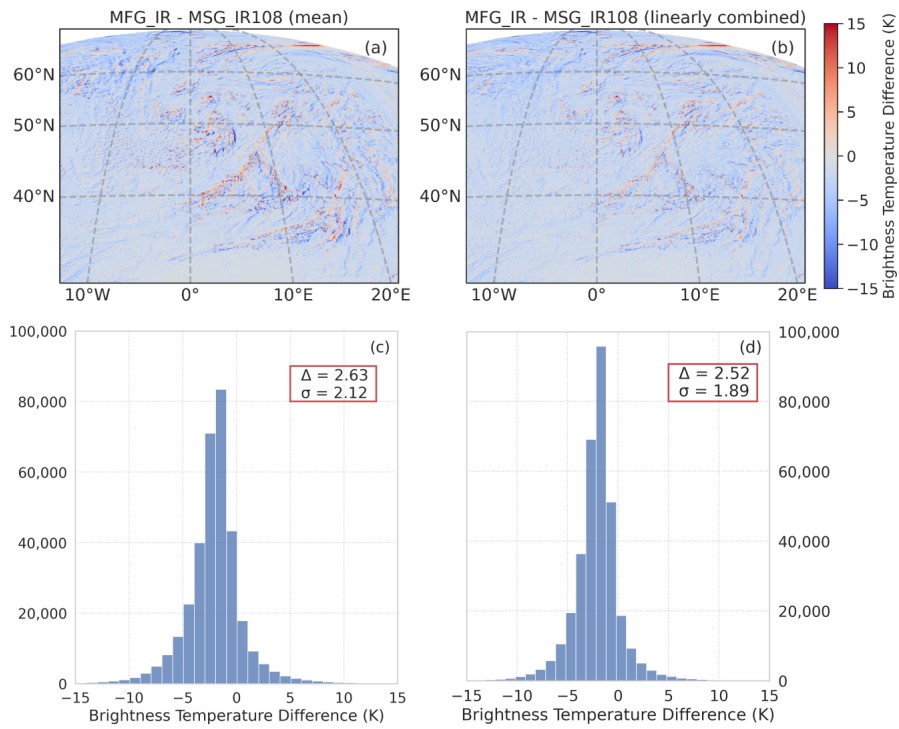

**Figure 6.** Difference between MFG original (IR channel) and temporally resampled MSG (IR 10.8 μm channel) scene for an example scene—1 January 2006, 09:00–09:30. (**a**) Difference with respect to the MSG scenes combined using mean BT values. (**b**) Difference with respect to the MSG scenes linearly combined (method adopted). (**c**) Histogram for difference shown in plot (**a**) with mean absolute shift (Δ) of 2.63 K and standard deviation (σ) of 2.12 K. (**d**) Histogram for difference shown in plot (**b**) with mean absolute shift (Δ) of 2.52 K and standard deviation (σ) of 1.89 K.

### 2.5. Machine Learning-Based Model for Inter-Calibration of MFG and MSG Datasets

2.5.1. Basic Overview of the Model

A machine learning (ML)-based model was trained on the overlap period (2004–2006) of both generation datasets to predict the MFG WV/IR channel BT values based on the MSG channels along with other predictors. For our study, we used a random forest (RF) regressor [30]. The reason for selecting an RF model over a multiple linear regression-based model is described in detail in the Section 2.5.3. RF is a supervised ML technique that uses an ensemble learning method. Each decision tree, a representation of a set of conditions that are organized hierarchically and applied sequentially from the root to leaf of the tree [31], in the RF model is trained independently by using a bootstrap sample from the training dataset. The various tree solutions are averaged at the end to predict the target variable with minimum error [32]. The ability of RF to evaluate only a subset of the original predictors at each split during the tree-building process and to parallelise the process (since the trees do not depend upon each other), makes it computationally efficient while working with large datasets. Additionally, it also provides the feature importance of the predictors which helps in selecting the relevant features. Due to these benefits of a RF model, it was chosen for the inter-calibration purpose. The RF regression models in this study were implemented using Scikit-learn which is a Python module incorporating a wide range of state-of-the-art machine learning algorithms [33].

In this study, two separate models, i.e., WV and IR models, were trained to predict and synthesize MFG WV and IR channels. The RF models were trained using the different channels combination of MSG along with other predictors to predict the BT values of MFG WV and IR channels. The MVIRI and SEVIRI data from the overlap period between Meteosat-7 and -8 in the years 2004–2006 were used to build and train the model.

2.5.2. Basic Model Input Features (MSG Channel Combinations)

Based on the preliminary investigation of the spectral response curves of Meteosat-7 (MFG MVIRI) and -8 (MSG SEVIRI), MSG WV062 and MSG WV073 channels were selected as the primary predictors for the WV model. Similarly for the IR model, three MSG IR channels (MSG IR108, MSG IR120 and MSG IR134) were chosen as the main input features. Figure 7 shows the MFG and MSG channels pairwise scatter plots of BT values of randomly selected pixels taken from the random MFG/MSG scenes of the overlap period (2004–2006). The scatter plots between MFG and MSG WV/IR channels provide an explanation of why these particular channel combinations are chosen for our model. The coefficient of determination ($R^2$) is used as an evaluation metric to assess the scatter of data points around the fitted regression line.

Due to significant spectral overlap between MFG WV and MSG WV062 and the proximity of their central wavelength, a high $R^2$ value of 0.97 can be observed. On the other hand, in the case of MFG WV and MSG WV073 channels the $R^2$ value (0.86) is not as high as that of MFG WV-MSG WV062 pair as the spectral overlap between the two channels is not quite significant (see Figure 4). This can be seen in terms of Mean Absolute Error (MAE), as well as with the values of 2.4 and 18.6 K for plot (a) and (b), respectively. However, MSG WV073 was also included as one of the input predictors in the WV model because of some spectral overlap with the MFG WV channel and a decent $R^2$ value. For the IR channels, the MSG IR108 and MSG IR120 channels have the maximum spectral overlap with the MFG IR channel which results in high $R^2$ value of 0.98 for each of these pairs, i.e., MFG IR-MSG IR108 and MFG IR-MSG IR 120. For the MFG IR-MSG IR134 pair, the $R^2$ of 0.95 represents a high level of correlation between the BT values of the two channels. The MAE for these three pairs are 2.8, 2.1, and 15.3, respectively. The best fit line equations in each of the scatter plots (Figure 7) can be used to obtain an idea of how close the BT values of the pixels of MFG-MSG channels pair are. Based on these scatter plots, the three MSG IR channels, i.e., MSG IR108, MSG IR120, and MSG IR134 were included as the input predictors for the IR model. The low *p*-values in the scatter plots also indicate that the independent variables have significance on target/dependent variables.

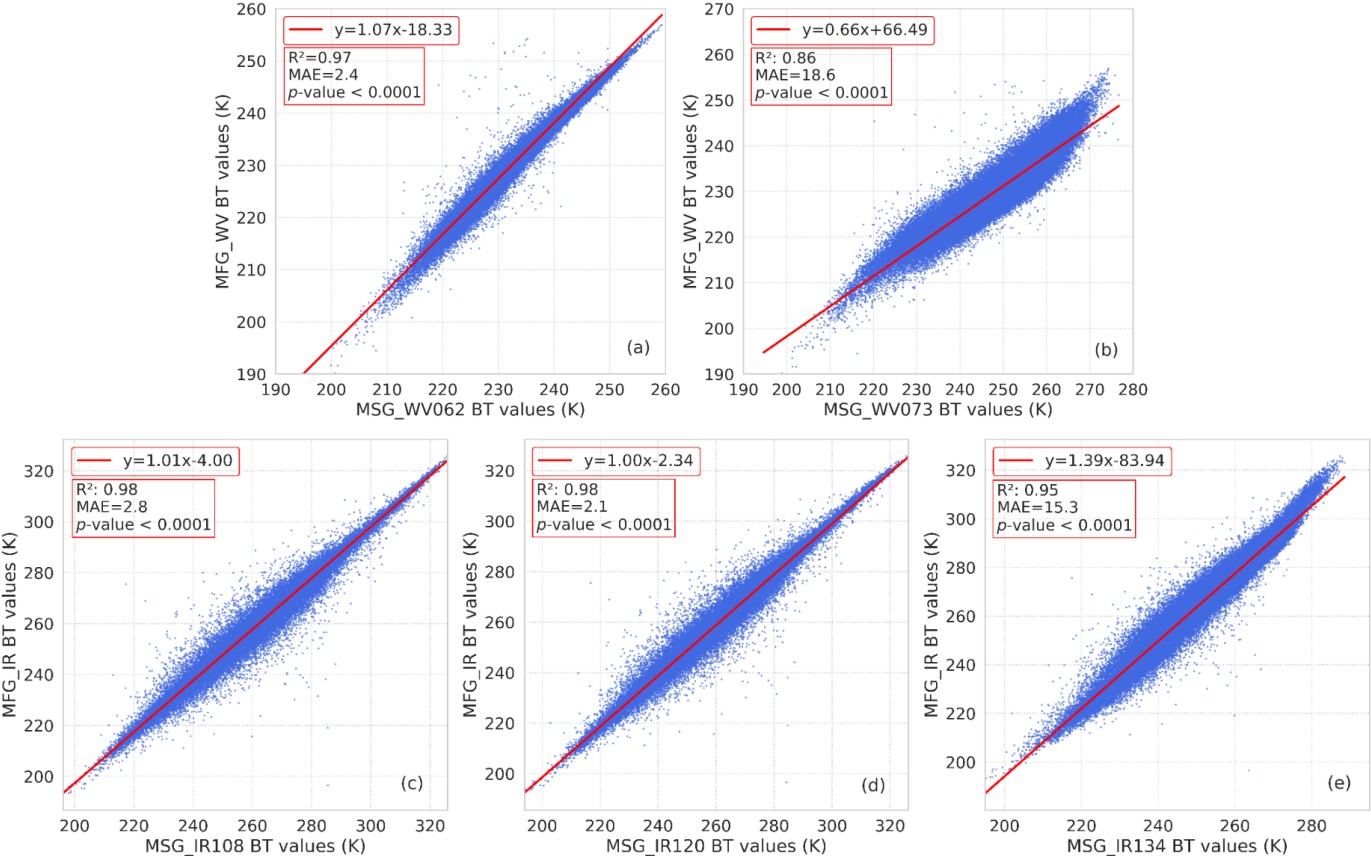

**Figure 7.** Scatter plots of MFG-MSG channels BT values (K). (**a**) MSG WV062 vs. MFG WV, (**b**) MSG WV073 vs. MFG WV, (**c**) MSG IR108 vs. MFG IR, (**d**) MSG IR120 vs. MFG IR, and (**e**) MSG IR134 vs. MFG IR. The red line in each scatter plot represents the best-fit regression line. These scatter plots were generated based on randomly selecting 100,000 MFG/MSG pixels from the overlap period (2004–2006).

### 2.5.3. Reason for Selection of RF Model over Multiple Linear Regression

Before selecting the RF models for this study, multiple linear regression based model were tested to synthesize the MFG WV and IR channels because this would be the easiest way of processing. Initially the models were trained just using the MSG channel combinations, as described in Section 2.5.2. When the synthesized scenes using this model set-up were compared with the original MFG scenes during the overlap period, an east–west trend was observed in the mean difference composite plot (Figure 8a,b). The red (hotter) region in the east represents the models under estimating the BT values. Similarly, the blue (colder) region in the west depicts the overestimation of BT values by the trained WV or IR model. This can be related to the different satellite viewing angles of the MFG and MSG sensors. To remove this effect, satellite viewing angles were added as the input predictors in the WV and IR models. The multiple linear regression-based WV and IR models were re-trained after adding the satellite viewing angles to the list of model input predictors. The satellite angles were clearly able to remove most of the east–west trend initially observed in the mean difference composites between the original and synthesized MFG scenes (Figure 8e,f). However still some colder–hotter pixel patterns can be seen especially for the WV model. These prediction results were further improved when the RF models were trained for the same training data and set of input predictors (Figure 8i,j).

To compare the different model settings in terms of their predictive power in computing the MFG WV and IR channel BT values, the mean difference composites based on the difference between MFG original and synthesized scenes were plotted and a histogram

analysis of these difference composites was performed. For the histogram analysis, the 5th (p5) and 95th (p95) percentile of the data distribution of the difference composites were considered and compared for each model setting. These values are shown in Table 1. p50 represents the 50th percentile or the median of the data distribution. For the first case (I), the range of values between p5 and p95 were −0.5 to 0.2 K and −0.5 to 0.5 K for the WV and IR channel difference composites, respectively (Figure 8c,d). The histogram analysis shows an improvement in the prediction results after adding the satellite viewing angles to the models (setting II) . The p5 and p95 values for the WV and IR difference composites were reduced to −0.3 to 0.1 K and −0.3 to 0.3 K, respectively. This is evident from the width of the data distribution in the histogram plots (Figure 8g,h). The performance of the RF models were even better in terms of synthesizing WV and IR channels which can be seen from the difference plots and the decrease in the width of data distribution of the histogram plots especially for the WV model (Figure 8k,l). For RF model setting (III), p5 and p95 were further reduced in the range of −0.1 to 0.0 K for the WV model and −0.2 to 0.2 K for the IR model. Based on this analysis, finally the RF models were selected in this study to harmonize the MFG and MSG scenes.

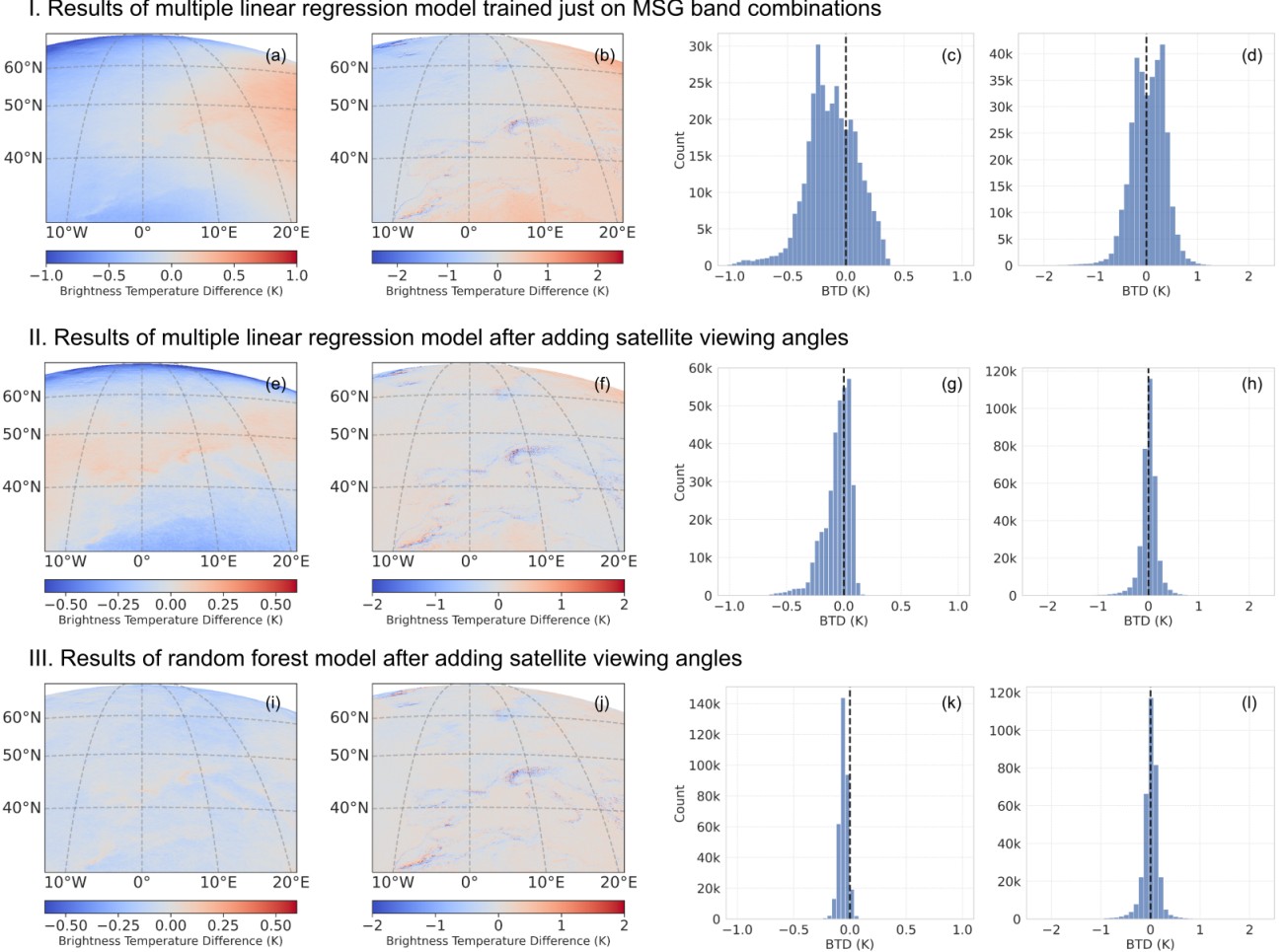

**Figure 8.** Mean difference composite plots generated based on the difference between MFG original and synthesized WV/IR channels and the histogram analysis of these composites for three different model settings. (**a**,**e**,**i**) show the mean difference composites for the WV channel and their histogram plots are shown in (**c**,**g**,**k**), respectively. (**b**,**f**,**j**) show the mean difference composites for the IR channel with their histogram plots represented by (**d**,**h**,**l**) respectively. The values represent the brightness temperature differences (K) between the original and synthesized scenes. These composites are generated by randomly selecting 2000 scenes from the overlap period (2004–2006).

**Table 1.** Percentile values of the histogram analysis for the three different model settings to train the WV and IR model shown in Figure 8. I, II and III represent the cases for different model settings as described in Figure 8. p5, p50 and p95 represent the 5th, 50th, and 95th percentile of data distribution in the histogram analysis for each case. The values are mean BT shift (K) between the original and synthesized MFG channels.

| | WV Channel | | | IR Channel | | |
|---|---|---|---|---|---|---|
| | **p5** | **p50** | **p95** | **p5** | **p50** | **p95** |
| I | −0.5 | −0.1 | 0.2 | −0.5 | 0.0 | 0.5 |
| II | −0.3 | −0.0 | 0.1 | −0.3 | 0.0 | 0.3 |
| III | −0.1 | −0.1 | 0.0 | −0.2 | 0.0 | 0.2 |

### 2.5.4. Final Model Input Features and Training Data

As described in the Section 2.5.3, the inclusion of the satellite viewing angles to the list of input predictors other than the MSG channels improved the performance of the models. The satellite elevation angle normally varies from 0 deg at the disk edges barely rising above the horizon to 90 deg at nadir. For our study region (30°–75°N), the satellite elevation varies approximately from 1° to 55°. Apart from the MSG channels and satellite viewing angles, the solar geometry (solar declination and zenith angles) parameters were added as predictor variables to the models in this study. Although the solar angles do not have any direct physical impact on the WV and IR channels, they can have an indirect influence on the brightness temperature of these channels. Due to different solar angles, differential heating in different latitudes and at different times of the year might have an indirect impact on the BT values of these channels. Sun Declination here is the angle between the sun ray extended to the center of the Earth and the equatorial plane, which mainly changes due to the rotation of the Earth about an axis. Instead of taking different times of the year into account by specifying the day or month, solar angles were considered in the model. While solar zenith angle is a measure of hour angle on a particular day, sun declination angle changes for each day of the year, varying from −23.5 deg to +23.5 deg in the course of the year. The complete list of input features used in WV and IR models are listed in Table 2.

**Table 2.** Overview of RF predictor variables (WV and IR models).

| | **MSG Channels** | **Satellite Angles** | **Solar Angles** |
|---|---|---|---|
| **WV Model** | MSG WV062 (6.2 μm) MSG WV073 (7.3 μm) | Azimuth Angle Satellite Elevation | Sun Declination Zenith Angle |
| **IR Model** | MSG IR108 (10.8 μm) MSG IR120 (12.0 μm) MSG IR134 (13.4 μm) | Azimuth Angle Satellite Elevation | Sun Declination Zenith Angle |

For the RF model creation, 3000 random MFG MVIRI and MSG SEVIRI scenes were selected from the overlap period (2004–2006). The randomly selected scenes account for different times of the day, as well as all the months of the year. The 3000 scenes were split into training and testing data. Out of the selected 3000 scenes, 2000 scenes were utilized for the purpose of training and 1000 scenes were kept for the independent validation, i.e., testing of the models. From each of these scenes, 2000 random pixels corresponding to the MFG/MSG channels of the predictand and predictor variables were selected and their BT values were extracted. This gave us ∼4 million data points for training and ∼2 million data points for testing the model. The training dataset was a further subset, with 70% of data points remaining in the training set and 30% in the validation set. The validation dataset was utilized to determine the optimal configuration of the models by hyperparameter tuning of the RF models.

Once the training and testing datasets were ready, the development of WV and IR models took place in three steps:

1. Model tuning: the hyperparameters of the RF models, such as the number of decision trees (n_trees) and the maximum depth of the tree (max_depth), were adjusted to obtain the optimal results.
2. Model training: the RF models were trained on the training data based on the optimal values of the model hyperparameters in the previous step.
3. Model validation: the trained RF models were tested on the separate testing datasets and model performance metrics were calculated.

All these steps are described in the following sections.

### 2.5.5. Model Tuning

Before training the WV or IR model, the two important hyperparameters, i.e., n_trees and max_depth were tuned. These two hyperparameters are important as they help to determine the computational burden incurred to train the model. As the number of decision trees and the maximum depth to which a tree can grow is increased, the model becomes slower with greater training time. Therefore, these hyperparameters can be considered a good stopping criteria when the performance of the model saturates. This was accomplished by an iterative approach with n_trees ranging from 50 to 300 and max_depth ranging from 5 to 30. The tuning dataset was fitted using RF models for each set of n_trees and max_depth, and the model's performance was analyzed on the basis of the computed mean absolute error (MAE). The results for WV and IR models are shown in Figure 9.

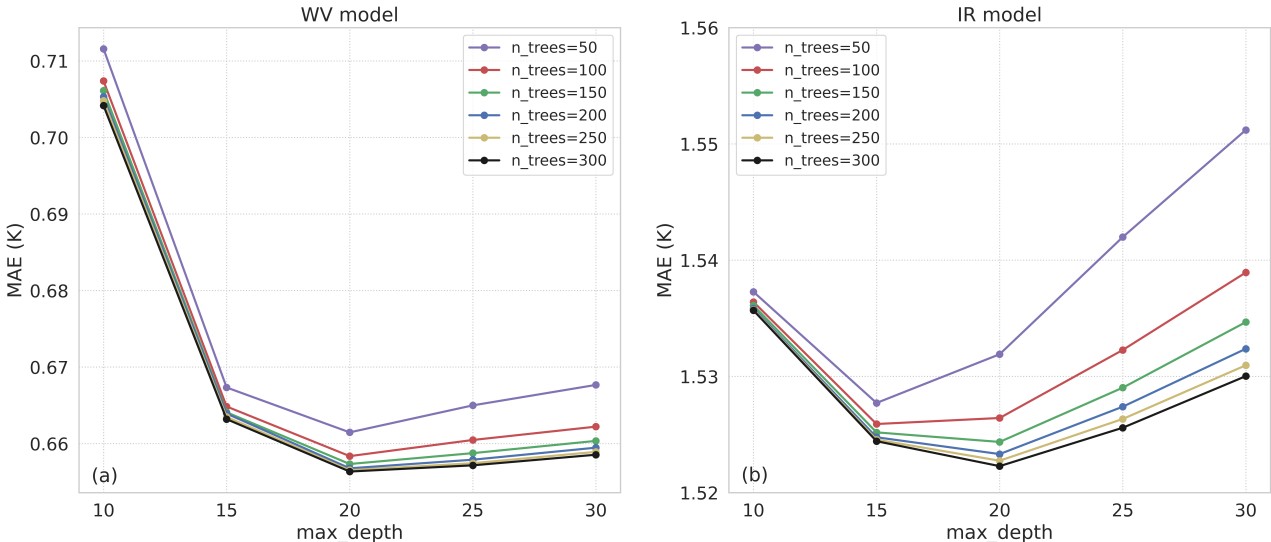

**Figure 9.** RF hyperparameters tuning results by visualizing the effect of number of trees (n_trees) and maximum depth of the tree (max_depth) on MAE. Results are shown for (**a**) the WV model, and (**b**) the IR model, respectively.

For the WV model, the MAE strongly decreased in the beginning up to max_depth = 15. The model starts stabilizing around max_depth = 20 and reaches its saturation level. As the maximum depth of the tree was increased further, it gradually began to start underperforming. The n_trees lines start converging around n_trees = 300 (see Figure 9a). As a result, the optimal values of max_depth and n_trees for the WV model were considered as 20 and 300, respectively, because adding more and more trees would have made the model computationally expensive without significant further improvement in the model performance.

For the IR model, the MAE decreased up to max_depth = 20 and started increasing again post that point. At max_depth = 20, the MAE starts to converge for 250–300 n_trees (see Figure 9b). So, 20 and 300 were selected as the optimal values for the max_depth and n_trees, respectively.

In the regression context, Breimann [30] suggests setting the number of randomly selected predictors at each split, commonly referred as $m_{try}$ to be one-third of the number of predictors. Since, the number of predictors for WV and IR models were 6 and 7, respectively, the optimal value of $m_{try}$ considered was 2.

2.5.6. Model Training

The RF regression models to synthesize the MFG WV and IR channel BT values were trained on a pixel basis using the randomly selected 2000 MFG and MSG scenes from the overlap period. From these 2000 scenes, a total of ~4 million pixels were selected for training the WV and IR models as described in the Section 2.5.4. Other predictors, such as satellite and solar angles (Table 2), were calculated for the selected pixels and finally fed to the WV and IR models. The RF models were trained using the optimal values of the model hyperparameters found in the Section 2.5.5. The RF model training, hyperparameter tuning, as well as the MFG data synthesis procedure in this study, were run on an Intel Core i7-7820X CPU Octa-core (8 Core) 3.60 GHz processor. For training both WV and IR models, it took ~15 mins for each model. The processing time to synthesize final MFG scenes were ~8 secs per scene.

2.5.7. Model Validation

Once the WV and IR models were trained, the models were validated by a separate set of testing data which consists of ~2 million pixels from the 1000 randomly selected scenes from the overlap period. The performance of the models was assessed by checking how accurate the model predictions were for the MFG WV and IR channel BT values. This was performed by calculating the regression models evaluation metrics, such as Mean Absolute Error (MAE) and Root Mean Square Error (RMSE). These metrics give the measure of amount of deviation of the model predicted values from the actual values. For WV and IR models, these metrics indicate the deviation of the predicted BT values ($BT_{pred}$) of MFG WV or IR channel from the actual BT values ($BT_{orig}$) of these channels. The *MAE* and *RMSE* were calculated as follows:

$$MAE = \frac{1}{N} \sum_{i=1}^{N} |BT_{orig} - BT_{pred}| \tag{8}$$

$$RMSE = \sqrt{\frac{1}{N} \sum_{i=1}^{N} (BT_{orig} - BT_{pred})^2} \tag{9}$$

Another metric used to evaluate the model performance was out-of-bag (OOB) $R^2$ score. The OOB samples are those which are not included in the bootstrap samples used to fit the decision trees [34]. The OOB score is often used to evaluate the predictive performance of the RF models. The OOB validation has the advantage of using the entire original sample for both RF model construction and error estimation. On the other hand, with cross-validation and associated data splitting approaches for error estimation, a subset of samples can be left out for the RF model construction [35]. Additionally, the computational speed is the another benefit of the OOB validation [36].

**3. Results and Discussion**

*3.1. RF Models Prediction Results*

Table 3 illustrates the performance of WV and IR models in terms of how well they predict the BT values (K) of MFG WV and IR channels. The deviation of the predicted BT values from the actual BT values of the respective channels is shown by different accuracy metrics as discussed in the Section 2.5.7. The MAE and RMSE for the WV model are 0.7 K and 1.0 K, respectively. For the IR model, the MAE and RMSE are 1.6 K and 2.7 K, respectively. The higher values of MAE and RMSE for the IR model can be attributed to the fact the range of BT values in the IR channel is larger as compared to that of the WV

channels since the WV channel only senses the radiation from the mid and upper levels of the atmosphere whereas the IR channel is used to determine the surface temperature. An OOB-$R^2$ of 0.98 for both the models indicate a good match of the MFG synthesized data with the original MFG data.

**Table 3.** Performance metrics of RF models (WV and IR models).

|  | MAE | RMSE | OOB-$R^2$ |
|---|---|---|---|
| **WV Model** | 0.7 | 1.0 | 0.98 |
| **IR Model** | 1.6 | 2.7 | 0.98 |

To compare the behavior of the RF regression models further, Figure 10a,b illustrates the scatter plots which compare the predicted and original BT values of MFG WV and IR channels. The plots placed horizontally at the top of the bivariate graphs show the distribution of the synthesized MFG WV/IR channel BT values. Similarly, the plots placed on the right margin of the bivariate graph with vertical orientation show the distribution of the original MFG WV/IR channel BT values. For this plot, a random 1 million pixels were selected from the random 2000 original/synthesized MFG scenes from the overlap period, i.e., 2004–2006. From the scatter plots, a good consistency can be observed between the original and predicted BT values for both the MFG channels (WV and IR), with the scatter points saturated around the 1:1 reference line at 45° (black line). The histograms showing the distribution of original and synthesized data (on the right and top respectively) looks similar based on the visual inspection, indicating similar data distribution for both. The mean ($\mu$) and standard deviation ($\sigma$) values for the original and synthesized datasets for both channels (WV and IR) are also shown in the Figure 10a,b which match perfectly well. This demonstrates the capability of both models in order to synthesize MFG WV and IR channels with reasonable accuracy.

Figure 10c,d displays box plots depicting the median and quartiles of the dataset consisting of a random set of 1 million pixels selected from the random 2000 original/synthesized MFG scenes from the overlap period to visualize the distribution of the data and to compare the BT values of the original and synthesized MFG WV and IR channels. The lower and upper limit of the box represents the 25th (lower quartile) and 75th percentile (upper quartile), respectively, with the median value shown by the horizontal line in the middle. This range of values, also known as interquartile range (IQR), is a useful measure of data dispersion because it is less influenced by the extreme values since it restricts the range to the middle 50% of the values. The lower and upper whiskers extend from lowest to highest BT values indicating variability outside the lower and upper quartiles. The data points outside this range can be considered as outliers. The red cross mark inside the box represents the mean of the datasets. The box plots indicate that the WV and IR models have performed well in predicting BT values of the MFG WV and IR channels with minimum, maximum, and quartile values matching each other (Figure 10c,d). The median values of original and synthesized WV and IR channels are the same for the set of 1 million pixels with their values as 230.4 K for the WV channel and 271.6 K for the IR channel. The results of the WV channel clearly indicate that the trained model is able to reduce the number of outliers significantly (Figure 10c).

### 3.2. Feature Importance

The variable importance, which indicates how much a given model uses that variable to make accurate predictions, is analyzed to determine the contribution of each input variable to the performance of the two models. The impurity reductions are summed over all split nodes in the tree to calculate variable importance in a tree-based regression model [37].

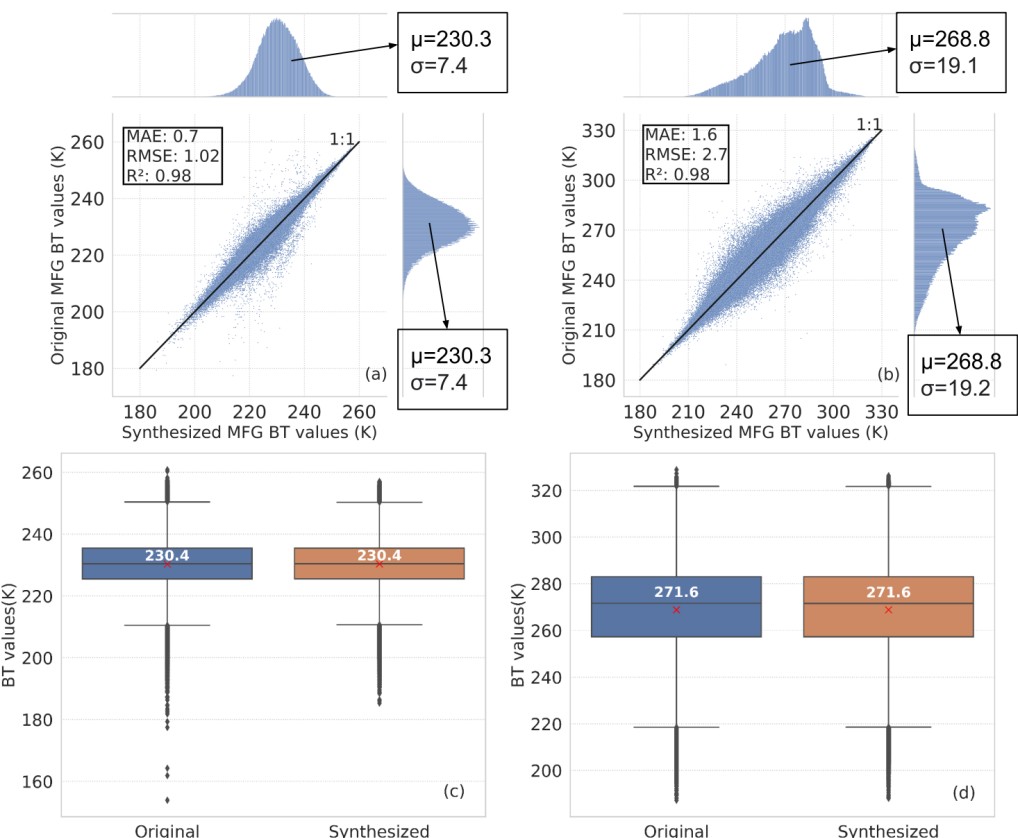

**Figure 10.** Evaluation of synthesized datasets—(**a**,**b**): Scatterplots depicting relationship between the synthesized and original values for WV and IR channel, respectively. The black lines in both the plots are 1:1 reference line. (**c**,**d**): Boxplots showing the distribution of the original and synthesized values for WV and IR channel, respectively.

For the WV model, the MSG WV062 has the highest contribution in synthesizing MFG WV channel with 54.1% contribution which is followed by MSG WV073 channel with 33.8% contribution (Figure 11a). The high contribution of the MSG channels can be explained by their spectral overlap with MSG WV062 channel having a significant overlap with the MFG WV channel (see Figure 4). The satellite elevation is another important input predictor with ∼10% contribution probably due to the longer atmospheric path over an absorbing medium which can impact the signals subjected to atmospheric absorption. Other input predictors have relatively less contribution with ∼1% or less feature importance, but they are kept in the model as these features helped in the better performance of the WV model. Similarly for the IR model, the most important features are the MSG channels (Figure 11b). The high contribution of MSG IR108 and MSG IR120 channels with 39.1% and 32.3% feature importance, respectively, can also be accounted for by the significant spectral overlap of these channels with the MFG IR channel (see Figure 4). Since the MSG IR134 channel has relatively less spectral overlap as compared to the other two MSG IR channels and it is on a $CO_2$ absorption band, its contribution is also less than these two channels with a feature importance of 22.9%. The satellite and solar angles also play a minor role in synthesizing the MFG IR channel.

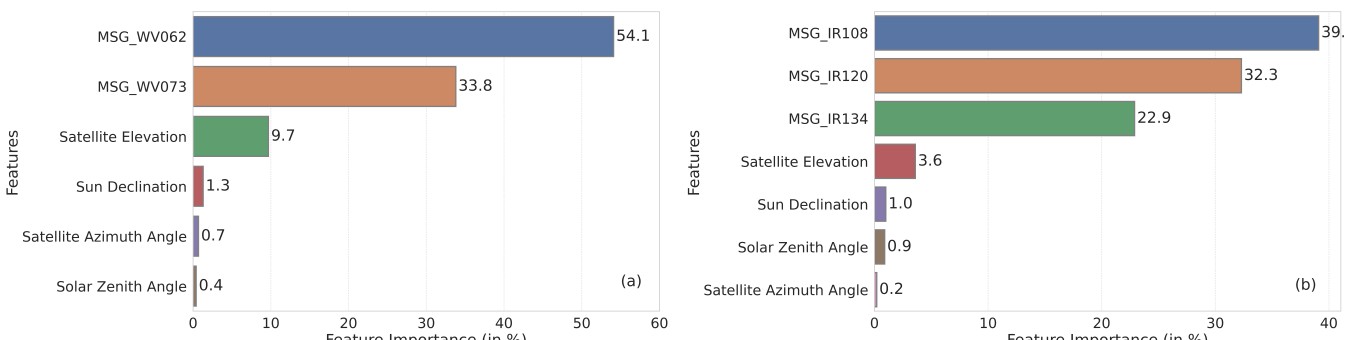

**Figure 11.** Feature importance for all the input predictors used in (**a**) WV and (**b**) IR model. The x-axis shows the feature importance (in %) and the y-axis shows the corresponding name of input variables.

### 3.3. Performance of WV and IR Models in Synthesizing MFG Datasets

Figure 12 shows how the harmonization of MFG and MSG datasets brought the two generation Meteosat datasets closer. The MFG and MSG mean composites were calculated based on randomly selected 2000 MFG and MSG scenes from the overlap period. The left side of the plot represents the mean composite differences between the original MFG and MSG channels. The MSG channels used here are spatially and temporally resampled to the resolution of MFG as described in Section 2.4. The right side of the plot shows the mean difference composite of the MFG original and synthesized channels. All the mean composite differences represent the mean BT shift in Kelvin.

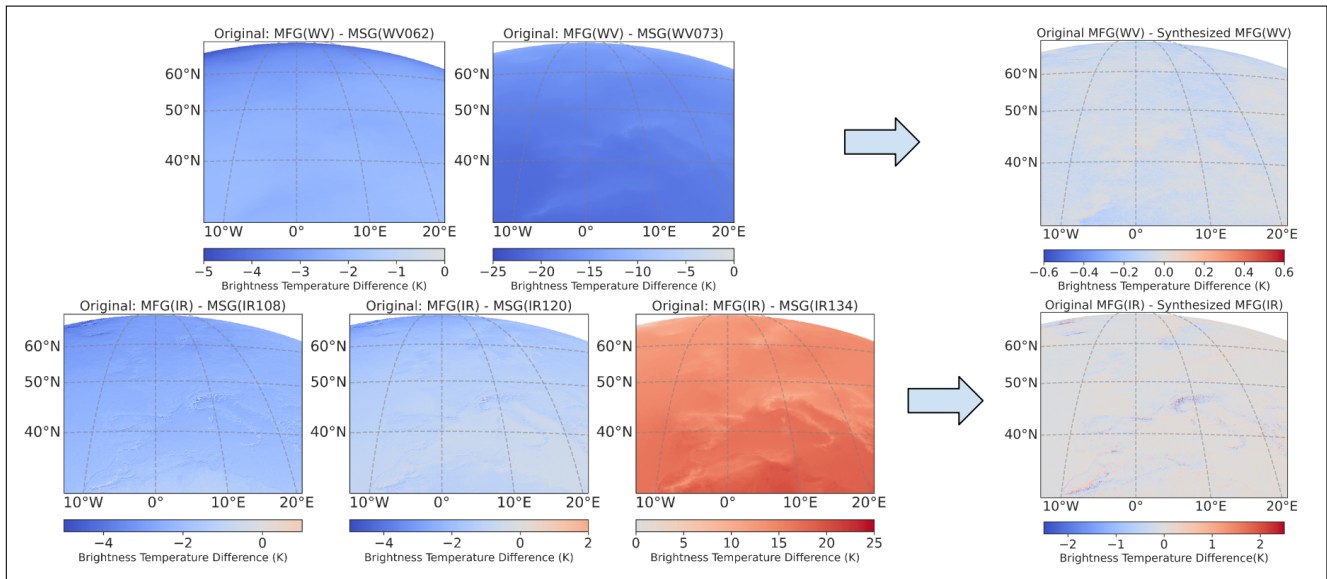

**Figure 12.** Mean composite differences between MFG and MSG original channels (**left**) and MFG original and synthesized channels (**right**). The top row represents the mean composite differences for the WV channels and the bottom row for the IR channels.

From the difference plots of the original MFG WV channel with respect to two MSG WV channels (MSG WV062 and MSG WV073) in Figure 12 (top row), expectedly a large shift of mean BT (up to 25 K for some regions) is observed for MFG WV and MSG WV073 difference. The reason for this large difference is that the WV absorption around 7.3 µm is not as strong as it is on the center of MFG WV absorption channel at 6 µm. The mean BT differences between MFG WV and MSG WV062 is relatively on the lower side due to the proximity of their central wavelengths. On the other hand, when the synthesized MFG WV channel BT values are compared with the original values for the same scenes by taking their mean composite differences, the differences seem to have decreased drastically

and now both BT values are very close to each other with the mean shift in the range of $-0.6$ to 0.6 K. Additionally, from the difference plot of the original and synthesized WV channel, no specific trend or pattern can be seen over the domain. When the mean composite differences are calculated for the original MFG IR channel with respect to the three MSG IR channels (MSG IR108, MSG IR 120, and MSG IR134), the BT values of MFG IR channel are on the lower side as compared to MSG IR108 and MSG IR120 channels with mean shift varying in the range of 0 to 5 K (see Figure 12 bottom row). A large positive shift can be seen in the difference plot of MFG IR and MSG IR134 channel indicating a much higher MFG IR BT values on an average in most of the regions of the study area. The 13.4 μm channel have a strong $CO_2$ absorption while 10 to 12 μm is almost transparent to the IR radiation. Here, the higher BT differences is because the lower cloud tops and surfaces are much colder on 13 μm due to the $CO_2$ absorption effect. The RF model based synthesized MFG IR scenes seem to reduce these gaps significantly which is evident from the difference plot of the original and synthesized MFG IR channel (Figure 12 bottom right). The model is overestimating and underestimating the BT values in some regions by 0–2 K. The result of the IR model seems to be pretty stable in almost all the parts of the European region except for some high-altitude areas. Some artifacts can be observed over some of the high-altitude regions, e.g., Alps, Pyrenees. This may have been caused by the apparent shift in one of the satellite's position during the overlap time period. The high-altitude pattern might have been because of the nearest neighbor resampling which can lead to the averaging effect of the lower MFG resolution becoming neglected and impose problems in the complex terrain when there is a high variability in MSG BT. To check for this, more complex resampling techniques were tried out but the same trend was observed in each case and no better result could be obtained by changing the resampling method. Apart from this unnatural artifact pattern, the mean BT shift in most of the regions are close to 0 K mark.

To understand the model's performance in more detail, the scenes were split based on different criteria, such as different range of elevations, satellite angles, and land–sea. This analysis was performed on randomly selected 1 million pixels from the overlap period. These pixels were further classified into different classes.

Elevation values for the study domain were extracted from the WorldClim DEM [28]. They were classified into four classes: low (<500 m), medium (500–2000 m), high (2000–3000 m), and very high (>3000 m). The performance of the WV and IR models in sub-regions classified in these four classes is shown in Figure 13a. The model's performance for both the models are almost the same for low- and mid-altitude regions. However, the mean deviation increases for higher altitude regions especially for the IR model. For very high-altitude regions (>3000 m), the mean deviation of synthesized IR channel BT values from the original is highest ($\sim$2.3 K). This can also be seen as the artifacts in high-altitude regions, such as the Alps and Pyrenees, in Figure 12. Apart from the apparent shift in one of the satellite's position, another reason for this can be the relatively fewer number of high-altitude pixels available for training the RF models. As a result, there might be a possibility that the IR model was not able to learn about the BT trends of high-altitude regions, as well as the other regions. This case is more evident for the IR model compared to the WV model because the IR channel measures the surface temperature unlike the WV channel. The satellite elevation angles were classified into three classes: low (<10°), medium (10°–45°), and high (>45°). Both WV and IR models appear to perform less well for low satellite angles towards the rim of scenes than for medium and high satellite angles with mean BT deviations $\sim$1 K and $\sim$2.6 K, respectively (Figure 13b). This may be due to the outliers in the original scenes along their edges. Another important factor for higher errors along the disk edges can be the changing pixel size growing larger towards the disk edges. Additionally, the higher absorption due to the longer atmospheric path on the disk edges might be one of the reason of the model's lower performance at low satellite elevation. No difference in the performance of WV and IR models is observed over land and sea pixels (Figure 13c). For the WV channels (especially the 6 μm channel), there is no

difference for land and sea pixels as the electromagnetic radiation emitted by land or sea is completely absorbed by the middle and lower tropospheric layers.

The month-wise parallel box plots representing original and synthesized WV and IR channels were plotted to obtain an insight of the performance of the trained models in different months (Figure 14). The parallel box plots indicate that both the models do a satisfactory job in estimating the WV and IR channel BT values throughout the year. Additionally, the WV model, in general, seems to be capable of reducing the frequency of outliers (Figure 14a).

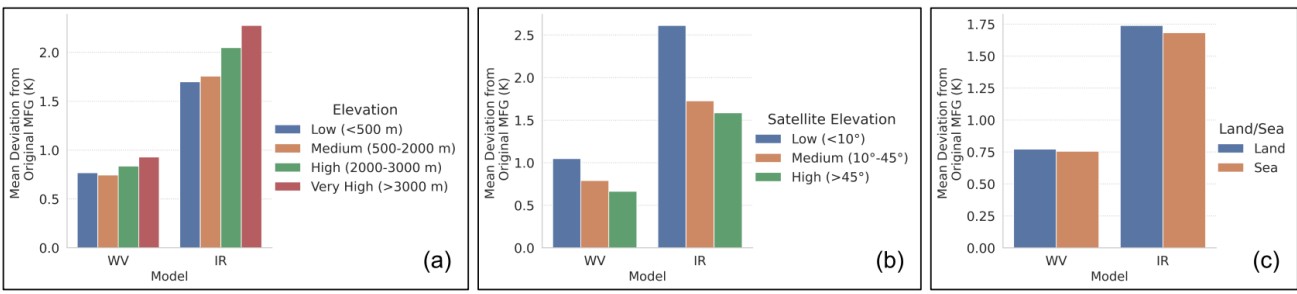

**Figure 13.** Performance of WV and IR models in different sub-regions classified according to: (**a**) Elevation, (**b**) Satellite Elevation (varies from 0° at the disk edges where satellite barely rises above the horizon to 90° at nadir), and (**c**) Land–Sea. The y-axis represents the mean deviation in synthesized MFG WV and IR channel BT values (K) from the original values.

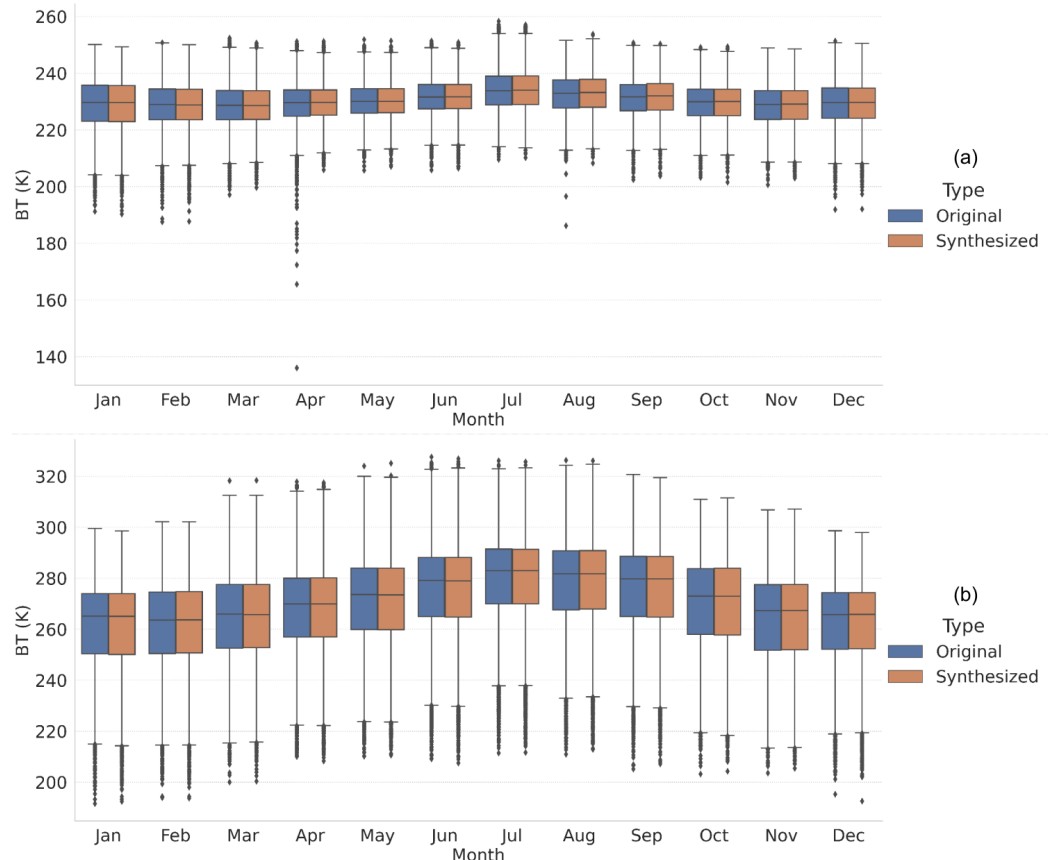

**Figure 14.** Monthly comparison of original and synthesized MFG scenes for (**a**) WV, and (**b**) IR channel. Box plots of the distribution of original vs. predicted BT values (K) for each month of the year. The boxes represent the 25th, 50th, and 75th percentiles. Whiskers extend 1.5 times the IQR to the most extreme data point (75–25th percentiles).

### 3.4. Homogeneity of Datasets

To check for the homogeneity of the generated dataset and original dataset from 1991 to 2020, year-wise time series plots of the MFG WV and IR channel mean BT values were generated to see if there are any discontinuities in the data (see Figure 15). The mean BT values for each year were calculated for four time slots in a day at a 6 h interval (0000, 0600, 1200, and 1800 h). The mean BT values in this analysis are for the region of interest of this study and not the full disk averages. To identify for any discontinuity in the data, it is very important to check for mean BT values whenever there is a change in Meteosat satellite. The checkpoints (red dotted lines in plots) represent these instances. For example, checkpoint I is placed in the year 1994 when the datasets received were from Meteosat-5 instead of Meteosat-4. Similarly, checkpoint II in the year 1997 represents the switch from Meteosat-5 to -6. The checkpoint IV marks the separation between the original and synthesized MFG data. Post this point, the datasets in WV and IR channels were synthesized from the trained models in this study. The detailed description of these checkpoints are given in Table 4.

**Table 4.** Description of the checkpoints placed in the time series plots (in Figure 15) to check the homogeneity of the long-term datasets. The black dashed line after checkpoint IV marks the separation between MFG original and synthesized datasets.

| Checkpoint | Year | Description |
|---|---|---|
| I | 1994 | Switch from Meteosat-4 to 5 |
| II | 1997 | Switch from Meteosat-5 to 6 |
| III | 1998 | Switch from Meteosat-6 to 7 |
| IV | 2006 | End of Meteosat-7 and the overlap period of Meteosat-7 (MFG) and Meteosat-8 (MSG) |
| V | 2013 | Switch from Meteosat-9 to 10 |
| VI | 2018 | Switch from Meteosat-10 to 11 |

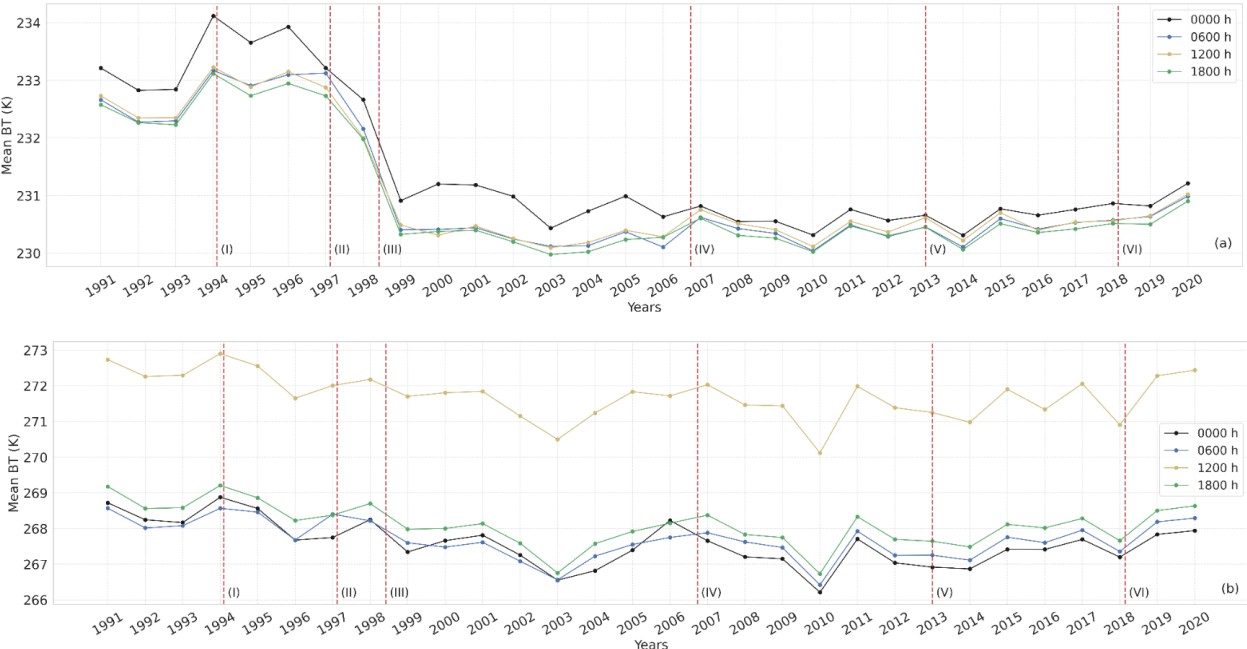

**Figure 15.** Time series plot (1991–2020) of mean BT values (K) for (**a**) WV and (**b**) IR channel. Red dotted lines represent the checkpoints (I to VI) placed whenever there is a change in Meteosat satellites (e.g., Meteosat-4 to -5 and so on) to check for homogeneity of the entire dataset by comparing the mean BT values before and after the change of satellite.

In order to examine the homogeneity around these checkpoints, 200 scenes were chosen at random for each timestamp slightly before and after each checkpoint, ensuring adequate coverage of scenes throughout the year and seasons. The respective means (μ) of the pixel BT values in these scenes are calculated and are listed in Tables 5 and 6.

**Table 5.** WV channel mean BT values (K) before and after the checkpoints.

| Check-Point | $\mu_{WV}$ (0000 h) | | $\mu_{WV}$ (0600 h) | | $\mu_{WV}$ (1200 h) | | $\mu_{WV}$ (1800 h) | |
|---|---|---|---|---|---|---|---|---|
| | Before | After | Before | After | Before | After | Before | After |
| I | 232.7 | 234.1 | 232.1 | 233.3 | 232.1 | 233.2 | 232 | 233.1 |
| II | 234 | 233.4 | 233.1 | 233.2 | 233 | 233 | 232.9 | 232.9 |
| III | 233.3 | 231.4 | 233.3 | 230.9 | 233.1 | 231.1 | 233 | 230.9 |
| IV | 230.8 | 230.6 | 230.1 | 230.3 | 230 | 230.5 | 230 | 230.3 |
| V | 230.4 | 230.3 | 230.3 | 230.1 | 230.2 | 230.3 | 230.2 | 230 |
| VI | 230.6 | 230.6 | 230.2 | 230.4 | 230.4 | 230.5 | 230.3 | 230.3 |

**Table 6.** IR channel mean BT values (K) before and after the checkpoints.

| Check-Point | $\mu_{IR}$ (0000 h) | | $\mu_{IR}$ (0600 h) | | $\mu_{IR}$ (1200 h) | | $\mu_{IR}$ (1800 h) | |
|---|---|---|---|---|---|---|---|---|
| | Before | After | Before | After | Before | After | Before | After |
| I | 267.9 | 269.2 | 267.7 | 269.6 | 271.7 | 273.4 | 268.1 | 270 |
| II | 268.7 | 267 | 268.3 | 268.5 | 272.2 | 272.1 | 268.7 | 268.4 |
| III | 268.1 | 268 | 268.9 | 268.4 | 272.7 | 272.9 | 268.9 | 269.1 |
| IV | 267.3 | 267.4 | 266.8 | 267.4 | 270.8 | 271.4 | 267.1 | 267.9 |
| V | 266.9 | 266.3 | 267.5 | 266.9 | 271 | 270.7 | 267.8 | 266.9 |
| VI | 267.5 | 266.8 | 267.4 | 267.6 | 271.7 | 271.5 | 268.1 | 267.7 |

The harmonized MFG datasets time series of mean BT values of WV and IR channels do not exhibit any prominent discontinuity between different Meteosat satellites. The only noticeable discontinuity can be seen in the WV channel time series, where there is a decrease in mean BT values by 2–2.5 K from the year 1998 to 1999 and the years onwards for all the timestamps used for this analysis. This shift is observed at the checkpoint III (Figure 15), when the Meteosat-6 was replaced by Meteosat-7. This may be caused due to the absence of a suitable on-board blackbody-based calibration system before Meteosat-7 [38]. Before Meteosat-7, the operational calibration relied on vicarious techniques for both the WV [39] and IR channel [40]. Earlier Meteosat satellites WV-channel calibration involved the method that combined radiosonde temperature and humidity profiles with clear-sky WV raw radiance (count) observations and radiances derived from a radiative transfer (RT) model [39,41]. The WV calibration needed continuous improvement because it was prone to bias errors due to incorrect radiosonde humidity measurements [42] which was used as an input for the RT model to compute radiances. This might be the reason contributing to the mean BT bias in earlier Meteosat satellites WV channel measurements before Meteosat-7 (Figure 15a). The time series plot post checkpoint III is very consistent for the WV channel with mean BT values in the range of 230–231 K (see Table 5). For the IR channel time series plot, no prominent discontinuity can be observed throughout the entire time span (1991–2020), although the difference between the mean BT values before and after the checkpoints are relatively on the higher side for the initial years as compared to the years post checkpoint III, i.e., 1998 (see Table 6). This may also be due to the calibration techniques involving the conventional meteorological data together with radiation model calculations in absence of an adequate on-board calibration. The range of BT values in the IR channel varies more between the different timestamps compared to the WV channel since it captures the surface temperature. Post checkpoint IV (the separation between original and synthesized datasets), a pretty consistent time series plot can be seen with no major fluctuations between different sensors and a similar trend of

plots for all timestamps. Overall, the results depict a very stable behaviour of the MFG WV and IR channels during the entire time period.

## 4. Summary and Conclusions

In the presented study, a robust workflow using the machine learning-based model is demonstrated for the harmonization of two Meteosat generation (MFG and MSG) datasets over the European domain (WMO region VI). For the harmonization process, the overlap period (2004–2006) was chosen where we have both MFG and MSG datasets available. The generation of the long-term harmonized dataset with high spatio-temporal resolution and extensive coverage over Europe in this paper is realized in four major steps: (i) prepro-cessing of MFG and MSG datasets received from EUMETSAT and bringing them to the common format; (ii) the spatial and temporal resampling of MSG datasets to the spatial (5 km) and temporal (30 min) resolution of the MFG datasets; (iii) creation and training of the RF models (WV and IR) to predict the BT values in MFG WV and BT channels using MSG channels, satellite and solar angles as the input predictors; and (iv) the generation of synthesized MFG datasets using the trained models (2006–2020). The spectral overlap between the MFG and MSG channels was considered as the main criteria for selecting MSG channels as input parameters in the models to synthesize the MFG channels. The trained models were validated against the original MFG scenes during the overlap period by comparing the predicted BT values with the original BT values in the MFG WV and IR channels. The predicted BT values in WV and IR channels seem to be in good agreement with the original BT values in these channels, with a MAE of 0.7 K for the WV model and 1.6 K for the IR model. The OOB-$R^2$ score for both the models was 0.98.

In this study, we have downgraded MSG datasets to MFG datasets in order to generate a long-term harmonized time series, not the other way around. Apart from the emissivity difference issue to derive the MSG 3.9 μm channel, the limited number of MFG channels are not sufficient to synthesize the complete information about MSG channels in general. For example, the spectral width of MFG IR channel is completely covered by the three MSG IR channels, i.e., IR10.8, IR12.0, and IR13.4, but the same cannot be said about the MSG channels. Another thing to note here is that the VIS channel was not included for the harmonization of MFG and MSG datasets. The reason for the exclusion of the VIS channel was to make the interpretation of properties of the diurnal cycle in the final product more feasible since the VIS channel is only available during day time. Although the focus has been put on synthesizing MFG WV and IR channels in this study because of their 24 h availability, it would be worth an effort in the future to build an RF-based model to simulate the MFG VIS channel from the MSG VIS channels because of its potential in different applications, such as solar radiation estimation studies, etc. The model's training data and input features can be chosen using the same approach used to build the WV and IR models in this study. Based on the spectral overlap of MFG and MSG, as shown in Figure 4, MSG VIS006 and VIS008 channels along with other features, such as satellite and solar angles, can be used to synthesize MFG VIS channel using the RF model.

The results of the RF models and homogeneity analysis show that the SEVIRI channels can be combined effectively to extend the MFG dataset 2006 onwards to present. After a thorough validation of the synthesized datasets with the original MFG datasets during the overlap period, the synthesized dataset from the year 2006 to 2020 was combined with the EUMETSAT FCDR of recalibrated Level 1.5 WV and IR radiances from the MVIRI instrument from 1991 to 2006 to create a long-term consistent MFG time series covering a climatological scale of 30 years. Due to stability issues and inhomogeneities in MFG datasets from MET-2 and MET-3 in the 1980s, these datasets were not included in the construction of this long-term harmonized dataset. This harmonized dataset based on the inter-calibration of MFG and MSG datasets is the first of its kind in terms of synthesizing the MFG datasets by exploiting the potential of a machine learning-based RF model. The RF model outperformed the standard linear regression-based model, particularly for the WV channel, as shown in the Section 2.5.3. Previous studies [14,15] mostly focused on

combining the two narrowband VIS channels of MSG satellites to simulate MFG broadband VIS channel. Due to the significance of WV and IR channels in FLS studies, the focus was put on simulating these channels in this study. In the models of this study, a few additional MSG WV and IR channels were included as input predictors based on their spectral overlap with MFG heritage WV and IR channels compared to other studies [19,21]. These additional MSG channels showed significant feature importance in synthesizing the MFG WV and IR channels. Apart from the MSG channels, the inclusion of satellite angles were also important in terms of removing the initially observed east–west trend in the mean difference composites. The WV and IR channels now synthesized with this study can be beneficial to conduct climatological studies, e.g., for fog and low stratus, as the synthesized dataset covers the climatological scale of 30 years (1991–2020).

The models have pretty stable behaviour throughout the European domain except for a few areas in the IR channel. When the synthesized MFG IR channel is compared to the actual MFG datasets, some artifacts can be noticed over some of the high-altitude regions, which may be caused by an apparent shift in one of the satellite's position during the overlap period or due to relatively less number of high altitude data points during the model training process. The performance of the models decreased to some extent along the rim of the scenes with low satellite angles, possibly due to the presence of outliers along the edge and the changing pixel size towards the disk edges. The predicted MFG WV channel BT values using the trained WV model in this study showed an improvement in terms of minimizing the number of data outliers present in the original MFG scenes. In general, the models seemed to perform very well in harmonizing the MFG and MSG datasets and bringing the two datasets closer. The long-term harmonized dataset generated in this study covering a time span of 30 years can be used for various climatological studies with a special care in the high-altitude regions, such as the Alps, and along the rim of the Earth due to relatively higher error in these regions. With the upcoming Meteosat Third Generation (MTG), a future extension of long term harmonized dataset will be a possibility using the machine learning based harmonization algorithm.

**Author Contributions:** Conceptualization and design of experiments: S.G., S.E. and J.B.; experiments performed: S.G.; data analysis: S.G., S.E., B.T. and J.B.; materials/analysis tools: S.E., B.T. and J.B.; paper writing: S.G. All authors have read and agreed to the published version of the manuscript.

**Funding:** This research was funded by the German Research Foundation (DFG) under the grants BE 1780/58-1 and EG 444/1-1. Open Access funding provided by the Open Access Publishing Fund of Philipps-University of Marburg with the support of DFG.

**Data Availability Statement:** All the processed and synthesized data are currently stored in the university network filespace and are available upon request by email to the corresponding author.

**Acknowledgments:** This study was performed under the project "SatFogClim: Is fog really decreasing everywhere?—A new long-term fog climatology for Europe based on cross-generation satellite data from the geostationary orbit" generously funded by the German Research Foundation (DFG) under the grants BE 1780/58-1 and EG 444/1-1. The authors are grateful to EUMETSAT for providing the MFG and MSG satellite data used in this study.

**Conflicts of Interest:** The authors declare no conflict of interest.

## Abbreviations

The following abbreviations are used in this manuscript:

| | |
|---|---|
| BT | Brightness Temperature |
| CFC | Cloud Fractional Cover |
| CM SAF | Satellite Application Facility on Climate Monitoring |
| COMET | Cloud Fractional Cover dataset from Meteosat First and Second Generation |
| DEM | Digital Elevation Model |
| EUMETSAT | European Organisation for the Exploitation of Meteorological Satellites |
| FCDR | Fundamental Climate Data Record |

| | |
|---|---|
| FLS | Fog and Low Stratus |
| HDF | Hierarchical Data Format |
| HRIT | High Rate Information Transmission |
| IR | Infrared channel |
| IQR | Interquartile Range |
| MAE | Mean Absolute Error |
| MFG | Meteosat First Generation |
| ML | Machine Learning |
| MSG | Meteosat Second Generation |
| MVIRI | Meteosat Visible and Infrared Imager |
| NetCDF | Network Common Data Form |
| OOB | Out Of Bag |
| RF | Random Forest |
| RMSE | Root Mean Square Error |
| SDECL | Sun Declination |
| SEVIRI | Spinning Enhanced Visible and Infrared Imager |
| SRF | Spectral Response Function |
| SZA | Solar Zenith Angle |
| VIS | Visible channel |
| WMO | World Meteorological Organization |
| WV | Water-Vapour absorption channel |

## Appendix A

The following figure shows why the temporal resampling technique based on the weighted average of the two MFG scenes was selected instead of selecting the nearest MSG scene. The difference plots show the higher differences if we consider the nearest MSG scene as compared to the MSG scene combined using our temporal resampling approach, as described in Section 2.4 for an exemplary MFG scene (30 October 2005, 13:00–13:30 h). The exemplary scene belongs to a day with a moving cloud system over the study region. This result holds true for checks performed on multiple time slots.

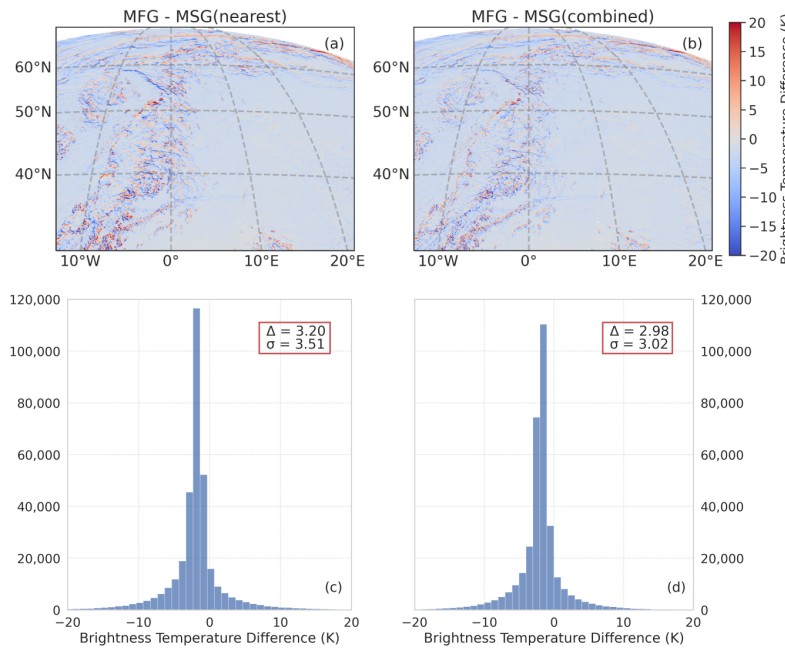

**Figure A1.** The difference plots of an exemplary (**a**) MFG IR original and the nearest MSG IR 10.8 µm scene, and (**b**) MFG IR original and two MSG IR 10.8 µm scenes combined using our approach, along with the mean (Δ) and the standard deviation (σ) values. Their respective histogram plots (**c**,**d**) are shown below in the figure. The exemplary scene belongs to a day with a moving cloud system over the study region (30 October 2005, 13:00–13:30 h).

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
