# Peer review of "Harmonization of Meteosat First and Second Generation Datasets for Fog and Low Stratus Studies"

_remotesensing, doi:10.3390/rs15071774_

Round 1

Reviewer 1 Report

The article is well written and structured, please see below few points that could be revised to make the text clearer.

199: There is a typo “...carried out by a the python...”

244: equations 5 and 6 – I think choose the nearest MSG image should be better than get a mean value of the before/after images. Clouds move enough in 15 minutes so a pixel representing a very cold cloud top in first scene can be an clear sky on the next scene. Maybe select only samples from MSG and MFG with brightness temperature difference smaller than few K in the window IR (~11 µm) would minimize the outliers.

504-505: add one sentence explaining why the satellite zenith angle is more important for WV synthesized images, probably due to longer atmospheric path over an absorbing medium.

511-513: add one sentence similar to one suggested above, 13 µm channel is on a CO2 absorption band.

525-527: I suggest to explain that the differences between MFG WV and MSG 7.3 WV channels are bigger because the WV absorption around 7.3 µm isn't stronger as is on the center of this WV absorption band at 6 µm.

540-541: 13.4 µm have a strong CO2 absorption while 10 to 12 µm is almost transparent to the IR radiation. Here, the higher BT differences is because the lower cloud tops and surfaces are much colder on 13 µm due to the CO2 absorption effect.

573-574: WV 6 µm images has no land or sea pixels, the EM radiation emitted by these objects is completely absorbed by middle and lower tropospheric layers, WV 7.3 µm can have some low cloud tops or high mountains in regions with strong subsidence. See the normalized weighting functions for WV MFG and MSG channels.

649-658: I suggest the authors to remove these sentences discussing why the 3.9 µm channel could be emulated from MFG channels.

662-665: Solar radiation estimative are done using these channels, would worth the effort.

697-701: The lower performance on the borders of the images could be the higher absorption due the longer atmospheric path?

Reviewer 2 Report

This manuscript presents the motivation, procedure, and validation of a random forest machine learning workflow to transform the higher temporal, spatial and spectral resolution data available from SEVIRI to an equivalent observation type provided by MVIRI. This dataset allows an examination of 30 years of IR data (window and water vapor channels) over the WMO region VI specifically for future study of the climatology of fog and low stratus clouds. 

Overall, this research was clearly presented which is useful for understanding the quality of this dataset and perhaps allowing other researchers to follow this approach. My background is in satellite data, not in Machine Learning, but I found all sections of this paper easy to follow with detailed explanations that I appreciated to have included. 

Questions and comments are all minor. 

1) The error presented is Mean Absolute Error. Was the bias or mean error also examined? Figure 10 does not give the linear regression intercept values for WV & IR results. It looks like there is no bias but I was wondering if that was true.

2) My convention for satellite zenith angles is 0 degrees for nadir and 70 deg near geostationary disk edge. Can you mention your convention for this angle when you first present it as a predictor? Perhaps add it to Fig 13 caption as well? You suggest higher errors at disk edge could be the source of the differences in Fig 13b. You did not discuss the changing pixel/field of view growing larger towards disk edge for both SEVIRI and MVIRI. I wonder if pixel size is also a factor?

3) The discussion of the topography impact on the window channel data uses the word noise in multiple places. To me, noise means random error. Would it be more accurate to call this an artifact or some other word? 

4) A hypothesis is suggested that the change in position of one of the satellites and/or small sample size of pixels for training may be responsible for the topography pattern. If a more complex averaging/sampling was used instead of nearest neighbor when switching to the coarser horizontal resolution, could this reduce the effect? 

5) Outliers are discussed and it is suggested that the lack of extreme outliers from the RF model is a good feature. I think you mean that the outliers have not impacted the model training in a negative way. Did you look at an example MVIRI scene with the very cold WV BT value to see if it was unrealistic or if it was a valid extreme measurement? If it was real and the RF model failed to predict it, would that still be a desirable quality for the RF model?

6) Section 3.4, maybe repeat that this is for the region of interest and not full disk averages. You could use the word discontinuity instead of jump. I appreciate the discussion about Fig 15a WV trends for the earliest sensors.

7) If a researcher was interested in following this approach, can you give any comments about what the RF model requires for computing resources? Both for hyperparameter tuning as well as running the workflow to create the dataset?

Reviewer 3 Report

The manuscript is well written and offers to the research community an interesting work on the harmozination of different meteosat datasets for application in the climate research of fog and low stratus. The study fills a gap and opens the field to several research works aimed at identifying the changes in the climate of the last three years starting from historical data of satellite homogenized. In our opinion the approach to the problem is right like the subsequent discussion of the results. The choice to use machine learning random forest instead of multiple linear regression is well explained and supported by the analysis carried. I think it is noteworthy the completeness of the literature review as well as the description of the used method and of the obtained results. Overall, my opinion is positive, I consider the work worthy to be published as it is. 

Author Response

Dear Reviewer,

Thank you for taking out the time and effort to review the manuscript.